# Initial Myeloid Cell Status Is Associated with Clinical Outcomes of Renal Cell Carcinoma

**DOI:** 10.3390/biomedicines11051296

**Published:** 2023-04-27

**Authors:** Saima Sabrina, Yuji Takeda, Tomoyuki Kato, Sei Naito, Hiromi Ito, Yuki Takai, Masaki Ushijima, Takafumi Narisawa, Hidenori Kanno, Toshihiko Sakurai, Shinichi Saitoh, Akemi Araki, Norihiko Tsuchiya, Hironobu Asao

**Affiliations:** 1Department of Immunology, Faculty of Medicine, Yamagata University, Yamagata 990-9585, Japan; shefa_610@yahoo.co.in (S.S.); s-saitoh@med.id.yamagata-u.ac.jp (S.S.); aaraki@med.id.yamagata-u.ac.jp (A.A.); asao-h@med.id.yamagata-u.ac.jp (H.A.); 2Department of Urology, Faculty of Medicine, Yamagata University, Yamagata 990-9585, Japanseinaito@med.id.yamagata-u.ac.jp (S.N.); uroushi@gmail.com (M.U.); tnari_0623@yahoo.co.jp (T.N.); ocean_ape@yahoo.co.jp (H.K.); tsakurai@med.id.yamagata-u.ac.jp (T.S.); ntsuchiya@med.id.yamagata-u.ac.jp (N.T.)

**Keywords:** CD16, glycosylphosphatidylinositol-anchored 80 kD protein, immune checkpoint inhibition, latency-associated peptide-1, myeloid-derived suppressor cells, renal cell carcinoma

## Abstract

The therapeutic outcome of immune checkpoint inhibition (ICI) can be improved through combination treatments with ICI therapy. Myeloid-derived suppressor cells (MDSCs) strongly suppress tumor immunity. MDSCs are a heterogeneous cell population, originating from the unusual differentiation of neutrophils/monocytes induced by environmental factors such as inflammation. The myeloid cell population consists of an indistinguishable mixture of various types of MDSCs and activated neutrophils/monocytes. In this study, we investigated whether the clinical outcomes of ICI therapy could be predicted by estimating the status of the myeloid cells, including MDSCs. Several MDSC indexes, such as glycosylphosphatidylinositol-anchored 80 kD protein (GPI-80), CD16, and latency-associated peptide-1 (LAP-1; transforming growth factor-β1 precursor), were analyzed via flow cytometry using peripheral blood derived from patients with advanced renal cell carcinoma (*n* = 51) immediately before and during the therapy. Elevated CD16 and LAP-1 expressions after the first treatment were associated with a poor response to ICI therapy. Immediately before ICI therapy, GPI-80 expression in neutrophils was significantly higher in patients with a complete response than in those with disease progression. This is the first study to demonstrate a relationship between the status of the myeloid cells during the initial phase of ICI therapy and clinical outcomes.

## 1. Introduction

Inflammation can play either a supportive or a suppressive role in cancer. Immunosuppression is often associated with chronic inflammation and is characterized by the presence of regulatory immune cells, including myeloid-derived suppressor cells (MDSCs), which can hinder antitumor immunity [1]. MDSCs suppress antitumor immunity by regulating cytotoxic T cells and natural killer cells [2]. However, MDSCs play a positive role in successful pregnancy [3]. MDSCs are also involved in resistance to angiogenesis inhibitors [4]. In recent years, targeting MDSCs has attracted attention as a new therapeutic modality for increasing the response rate to cancer therapy. Although the MDSC population is speculated to be primarily composed of a subset of immature myeloid cells resulting from unusual myeloid differentiation [5], a comprehensive analysis of MDSCs is required.

To improve the therapeutic outcome of immune checkpoint inhibition (ICI) for the treatment of various types of cancer, previous studies have attempted to combine ICI therapies with multiple treatments, such as molecular targeted therapy, radiation, and biologics [6]. However, these studies have not focused on analyzing the dynamics of the immune status of each patient, specifically the status of the myeloid cells. However, as the myeloid cell population consists of an indistinguishable mixture of various types of MDSCs, activated neutrophils, and monocytes, analyzing the status of the myeloid cells is challenging.

Advanced renal cell carcinoma (RCC) is resistant to conventional chemotherapy, radiotherapy, and hormonal therapy owing to its heterogeneity. Since 2016, ICIs have been introduced in Japan for the treatment of RCC, improving the overall survival period of patients by up to 27 months [7]. Although ICI therapy is associated with longer overall survival, it has low response rates (40%) in patients with RCC, with only 9% of patients showing a complete response [8].

Previously, we demonstrated a method for the identification of myeloid cells from peripheral blood via staining with the myeloid cell marker CD33 and the neutrophil maturation marker CD16 (FcγRIII) in cases of RCC [9,10,11]. We revealed that the expression of CD16 and the TGFβ precursor molecule latency-associated peptide-1 (LAP-1) on monocytic cells (CD33^hi^ cells) is an indicator of MDSC status, as these parameters are involved in the suppression of antitumor immunity in patients with RCC [12]. Moreover, we identified that increased expression of glycosylphosphatidylinositol-anchored 80 kD protein (GPI-80) is an indicator of neutrophil maturation [10]. Notably, elevated GPI-80 expression has been reported in activated neutrophils [13]. We further demonstrated that increased expression of the coefficient of variation (CV) of GPI-80 (GPI-80 CV) on neutrophilic cells (CD16^hi^ cells) indicates unusual differentiation of neutrophils related to MDSC functions [10,12].

In this study, we investigated the relationship between changes in the status of the myeloid cells and the therapeutic efficacy of ICI to provide fundamental findings for the application of ICI therapy. We examined changes in the myeloid cell parameters, CD16, LAP-1, and GPI-80, which have been proposed as MDSC indexes. As pro-inflammatory factors induce MDSCs [14,15], we estimated the levels of circulating cytokines and damage-associated molecular patterns (DAMPs) in patients with RCC. The myeloid parameters proposed in this study are useful for inferring the therapeutic efficacy of ICI.

## 2. Materials and Methods

### 2.1. Peripheral Blood Collection and Manipulation

This study was approved by the Ethics Committee of the Yamagata University Faculty of Medicine (approval no. H29-15). This study included 51 patients diagnosed with RCC in the Department of Urology at Yamagata University Hospital between May 2020 and September 2021. Peripheral blood (5–10 mL) was collected from patients with RCC using a heparinized collection tube after obtaining informed consent. Blood was collected immediately before and during treatment. These samples were maintained at 20–25 °C and utilized for experiments within 24 h. Detailed characteristics of the patients are summarized in Table 1. After the end of treatment, the best responses of 50 out of 51 cases were classified as complete response (CR), partial response (PR), stable disease (SD), and disease progression (PD) based on computerized tomography scan results using the response evaluation criteria in solid tumors (RECIST) guidelines (version 1.1), as shown in Table 2. The response of one patient was not evaluated (NE).

### 2.2. Flow Cytometry Analysis

Whole blood cells were stained with antibodies, as described previously [9,11]. Briefly, whole blood was aliquoted into microtubes (100 μL of blood per tube) and incubated with an Fc Blocker (BioLegend, San Diego, CA, USA) for 5 min. After blocking the Fc receptors, the whole blood was incubated with each antibody for 30 min at 4 °C and then treated with pre-warmed BD Phosflow Lyse/Fix buffer (1 mL; BD Biosciences, San Jose, CA, USA) for 10 min at 37 °C to lyse the red blood cells and fix the white blood cells (WBCs). After a washing with PBS, the cells were analyzed by flow cytometry using the FACSCanto II (BD Biosciences). The antibodies used in this study were as follows: PE-conjugated an-ti-CD16 mAb (3G8) and APC-conjugated anti-HLA-DR mAb (G46-6) from BD Biosciences; Brilliant violet 421-conjugated anti-CD33 mAb (WM53) from BioLegend; APC-conjugated anti latency-associated peptide-1 (LAP; the N-terminal region of transforming growth factor-β1 precursor) mAb (#27232) from R&D Systems (Minneapolis, MN, USA); and PE- or FITC-conjugated anti-GPI-80 mAb (3H9) from MBL (Nagoya, Japan). For isotype-matched control mAbs, IgG1 (MOPC-21) and IgG2a controls (G155-178) were obtained from BD Biosciences. The mean fluorescence intensity (MFI) and robust CV were analyzed using FlowJo software version 7.8.6 (TreeStar, Ashland, OR, USA) [10].

### 2.3. Measurement of Pro-Inflammatory Factors in Plasma

Plasma was separated from whole blood and centrifuged for 3 min at 2400× *g* and stored at −80 °C. The presence of various inflammatory cytokines, such as chemokines, growth factors, and interleukins (pg/mL), from 50 μL plasma samples of patients with RCC were analyzed using the LEGENDplex™ Human Inflammation Panel 1 (BioLegend, San Diego, CA, USA) and BD™ Cytometric Bead Array System (CBA) (BD Biosciences, San Jose, CA, USA) according to the manufacturer’s instructions.

High-mobility group box 1 (HMGB1) (ng/mL), one kind of DAMP, was further analyzed in the plasma samples of patients with RCC using the Human HMGB1 ELISA Kit (Arigo Biolaboratories, Taiwan, ROC) according to the manufacturer’s instructions.

### 2.4. Statistical Analysis

All statistical analyses were performed using EZR version 1.35 (Saitama Medical Center, Jichi Medical University, Saitama, Japan) and GraphPad Prism version 8.4.3 (GraphPad Software, San Diego, CA, USA). EZR is a graphical interface for R (R Foundation for Statistical Computing, Vienna, Austria). This is a modified version of R Commander version 2.3 [16]. The statistical processing methods are described in the figure legends. *p*-values less than 0.05 were considered to be statistically significant. The statistical methods used for the analyses are described in the figure legends.

## 3. Results

### 3.1. Patients’ Clinical Outcomes

Immune checkpoint inhibitors (anti-PD-1/PD-L1 and anti-CTLA-4) and other anti-cancer drugs were administered to patients with RCC. Patients treated solely with immune checkpoint inhibitors (ICIs) received either Nivolumab plus Ipilimumab (N + I) combination therapy or Nivolumab (N) monotherapy. Patients treated with ICI and tyrosine kinase inhibitors (TKIs) received Avelumab plus Axitinib (A + A), or Pembrolizumab plus Axitinib (P + A). Patients treated with other chemotherapeutic drugs received TKIs, including Sorafenib (SO), Cabozantinib (CABO), Pazopanib (PAZ), Axitinib (AX), fluoropyrimidine class chemotherapy TS-1, or the mTOR inhibitor Everolimus (EV). To clearly evaluate the immune responses to anti-cancer therapy, we divided the clinical outcomes of all patients into two categories: immunoresponse-dominant, including CR and PR, and immunoresponse-non-dominant, including SD and PD. The results are summarized in Table 2. In this study, 19 patients received ICIs as a first-line therapy (1st line). Patients who did not respond to the first therapy (1st line) were administered second-line therapy (2nd line), and if the treatment failed again, the next treatment was administered stepwise (3rd–6th lines). Although only 3 of the 20 responding cases were classified as CR, this ratio was almost consistent with previously reported rates [8,17]. As shown in Appendix A, there was no association between the TNM grade and clinical outcomes in patients treated with ICI therapy as a first line, while CR cases had histologically diverse RCC subtypes, as opposed to PR, SD, and PD cases, who had mostly clear cell RCC.

### 3.2. Changing Patterns of Myeloid Cell Parameters during ICI Therapy

We examined the myeloid cell parameters to determine the common changing patterns of myeloid cells immediately before and during ICI therapy. Representative blood collection during ICI therapy is shown in Figure 1a, and the gating strategy for the flow cytometric analysis of peripheral blood samples is shown in Figure 1b and Appendix A, according to previous reports [10]. Interestingly, when we focused on the expression levels of CD16 and LAP-1 in monocytic cells (Figure 2a,b) and the expression level of GPI-80 in neutrophilic cells (Figure 2c), these levels were not stable immediately before or during ICI therapy as a first line, even within separate clinical outcomes (CR, PR, SD, and PD). We also analyzed the myeloid cell parameters of patients treated with other therapy lines and regimens, but these results were similar to those of ICI therapy as a first line. These observations indicate that the changing patterns of CD16, LAP-1, and GPI-80 expressions vary depending on the patient.

### 3.3. CD16 and LAP-1 Increase in Immunoresponse-Non-Dominant Patients, but Not in Immunoresponse-Dominant Patients

Next, we separated the data into two categories of clinical outcomes (immunoresponse-dominant and immunoresponse-non-dominant) for all therapies. In immunoresponse-dominant patients (CR and PR), there was no change in the CD16 and LAP-1 expression in the CD33^hi^ monocytic cells from the first blood samples (before treatment) compared to those of the second blood samples (after the first treatment), since the expressions were variable in these patients (Figure 3a). In contrast, the CD16 and LAP-1 expression was slightly but significantly elevated in immunoresponse-non-dominant patients (SD and PD) (Figure 3b). Previously, CD16 expression was correlated with LAP-1 expression in CD33^hi^ monocytic cells, and augmentations of these expressions were detected in patients with metastatic RCC, compared to those in healthy volunteers [10]. These observations suggest that the slight increase in the CD16 and LAP-1 expression in the initial phase of cancer therapy reflects chronic inflammation associated with the cancer progression.

### 3.4. A High GPI-80 Expression Level in Neutrophilic Cells Associates with an Antitumor Immune Responses

Next, we focused on patients with a complete response to investigate whether antitumor immune responses were associated with changes in the myeloid cells. In this study, three patients (approximately 6%) who received ICI therapy as a first line showed CR. In order to compare with CR cases, patients who received ICI therapy as a first line were selected, as shown in Table 2 and Appendix A. Although we investigated many parameters of myeloid cells in peripheral blood using flow cytometry (Appendix A), the only characteristic parameter of CR in this study was the expression level of GPI-80 (GPI-80 MFI) in CD16^hi^ neutrophilic cells in the first blood sample immediately before ICI therapy. To clarify the change in myeloid cell status in RCC, previous results from healthy donors (HD) [10] are also presented in Figure 4. As described in our previous study, GPI-80 CV, which indicates variations in neutrophilic maturation, was lowest in HD patients, compared to patients with RCC with CR, PR, SD, and PD (Figure 4a). However, there was no difference in GPI-80 CV between the CR, PR, SD, and PD groups (Figure 4a). Although the GPI-80 MFI in HD patients was not different from that in PD patients, there was a significant difference in GPI-80 MFI between CR and PD patients (Figure 4b). These results suggest that the neutrophil state associated with GPI-80 expression immediately before ICI therapy is linked to the anti-cancer immune responses induced by ICI therapy.

### 3.5. Variation in Pro-Inflammatory Factors in the Plasma of Patients with RCC

Pro-inflammatory cytokines are useful for predicting immune responses to ICI therapy [18]. Furthermore, DAMPs are known to modulate the tumor microenvironment via MDSCs [15]. To validate the effect of pro-inflammatory factors on clinical responses, we measured these pro-inflammatory factors in the first blood samples of patients with CR and PD and receiving ICI therapy as the first line, as described in Table 2. Heat map analysis showed that there was no specific pattern of the 16 cytokine levels in the plasma of patients with CR or PD (Figure 5a). Similarly, no significant difference was found in the plasma HMGB1 concentrations between patients with CR and those with PD (Figure 5b). We also analyzed these pro-inflammatory factors in the second blood sample (after the first treatment) from the same patients to confirm whether there was any relationship between cytokine levels and the clinical response (Appendix A). However, no definite pattern was observed in the second blood sample (after the first treatment). Previously, it was reported that intratumoral cytokine levels were distinct from those of circulating plasma cytokines in patients with RCC [19]. These observations suggest that pro-inflammatory factors in plasma do not mirror the anti-cancer immune responses induced by ICI therapy.

## 4. Discussion

In this study, the myeloid cell parameters (CD16, LAP-1, and GPI-80) were unstable during ICI therapy and varied depending on the patient. The CD16 and LAP-1 expression in monocytic cells increased in the second blood samples (after the first treatment) of the patients who were immunoresponse-non-dominant to cancer therapies. Interestingly, CRs to ICI therapy were associated with GPI-80 upregulation in neutrophilic cells in the first blood sample that was collected immediately before treatment. These findings indicate that the initial status of the myeloid cells in patients with RCC is associated with the clinical response to ICI therapy.

Inflammation is an essential host-defense mechanism that eliminates harmful stimuli and restores tissue homeostasis. Anti-cancer therapy often induces immunosuppressive inflammation, which promotes cancer by blocking antitumor immunity [20]. This pro-tumorigenic inflammation leads to the unusual differentiation of myeloid cells and the consequent appearance of suppressive MDSCs [1]. Owing to the morphological resemblance of MDSCs to neutrophils and monocytes, various markers have been used to predict their appearance [21]. S100A9, a member of the S100 family of calcium-binding proteins, has been used as a marker to identify monocytic MDSCs [21,22]. S100A9 is expressed in granulocytes and monocytes and in the early phase of differentiating macrophages. It is also expressed in other cells, including keratinocytes and epithelial cells, during inflammation. Therefore, differentiating MDSCs from other S100A9-expressing cells under cancer-induced inflammatory conditions is difficult. Moreover, S100A9 expression is not an independent marker for the differentiation of monocytic MDSCs from monocytes. In contrast, lectin-like oxidized LDL receptor-1 (LOX-1) has been manifested as a useful marker of human neutrophilic MDSCs [3,21], although different MDSC subsets have been reported under different pathological conditions [23]. In our study, we performed LOX-1 staining and found no remarkable LOX-1 expressions in the peripheral blood of patients with RCC. Inflammatory conditions resulting from cancer progression can produce various subsets of MDSCs. Furthermore, the specific genomic and proteomic signatures that characterize these different MDSC subsets have not yet been validated [3]. In this situation, our proposed MDSC markers are useful for recognizing the heterogeneity and appearance of different MDSC subsets in the peripheral blood of patients with RCC using a simple staining method.

To completely understand the immune response to therapy, efficacy outcomes can be categorized into two different response patterns: objective responses, including CR and PR, and disease controls, including CR, PR, and SD [17]. In our study, immunoresponse-dominant categories (CR and PR) comprised the objective responses. The non-dominant immune response category included patients with SD and PD who were classified as experiencing a low immune response. Because this study was conducted with a small sample size, we particularly focused on objective responses to clarify the immune response after anti-cancer therapy.

Usually, tumors with high-cytokine expression have increased quantities of other immune cells, exclusively in the myeloid compartment, which contribute to the immune response. We measured the concentrations of various human inflammatory cytokines, such as- IL-1β, INF-α2, INF-γ, TNF-α, MCP-1 (CCL2), IL-6, IL-8 (CXCL8), IL-10, IL-12p70, IL-17A, IL-18, IL-21, IL-23, IL-33, G-CSF, GM-CSF, and HMGB1 in the plasma of patients with RCC. IL-6, TNF-α, G-CSF, and GM-CSF induce MDSC differentiation [1,14,24]. IL-1β and IL-8 are involved in the recruitment of MDSC in patients with RCC [25,26]. It has been reported that MCP-1 induces MDSC accumulation [27], whereas INF-α2 suppresses MDSC accumulation [28]. Plasma levels of INF-γ and IL-10 are found to be correlated to immune-related adverse events after ICI therapy [29,30]. IL-12 and IL-23, through innate immune cells and IL-33, by suppressing MDSC, potentiate immune responses during infection and cancer [31,32]. Several studies have reported that IL-17A increases PD-L1 expression and promotes resistance to ICI therapy [33,34]. IL-21 enhances neutrophil functions while playing key roles in antitumor and antiviral responses [9,35]. HMGB1 mediates tumor immune suppression by recruiting MDSCs, activating regulatory T cells, and inhibiting dendritic cell infiltration. All of these events are expected to result in poor outcomes of cancer patients after immunotherapy [36,37,38]. However, the immunoregulatory roles of proinflammatory factors are evident after ICI therapy in most cases, whereas we mainly investigated these factors immediately before and during the initial period of ICI therapy. Moreover, a previous report showed that plasma cytokine levels were not similar to intratumoral cytokine levels [19]. Consistent with this report, we observed no specific patterns in plasma cytokine levels in patients with RCC immediately before and during the initial treatment period. This result indicates that the plasma levels of these cytokines are not useful for predicting therapeutic efficacy in individual patients undergoing ICI therapy.

We aimed to determine the relationship between increases in GPI-80 MFI and GPI-80 CV in neutrophilic cells. Previously, we studied the relationship between cell behavior and MFI/CV patterns [39]. The increases in both MFI and CV indicate that they are “Type 2 (subsequent)” molecules. The term “subsequent” means that the function of the molecules, which was measured as Type 2, is transitioning to other phenomena after activation. Thus, the status of neutrophils in patients with CR immediately before therapy may indicate a transition to antitumor neutrophils from homeostatic neutrophils. In the future, molecules measured as “Type 2” may be useful for the prior detection of antitumor responses, similarly to GPI-80.

RCC is a histologically heterogenous cancer with most the common subtype being clear cell RCC (ccRCC) and all subtypes other than clear cell being categorized as non-clear cell RCC (nccRCC). In this study, only three cases of CR were reported with both ccRCC and nccRCC. Patients with nccRCC have limited treatment options, with the tendency of getting worse. Multiple clinical trials are now using ICI therapy for patients with nccRCC, though the effectiveness of ICI therapy in patients with nccRCC is not clearly understood yet [40]. Since patients with nccRCC showed better responses toward ICI therapy in this study, we will continue the investigation with both histological subtypes. The number of CR cases reported in this study is insufficient to make a concrete decision to use GPI-80 as a predictor of CR. Further studies are needed to validate the usefulness of GPI-80 in predicting therapeutic effects during the early period of treatment. However, this is the first study to report that there is a relationship between the state of the myeloid cells immediately before the treatment and the outcome of the ICI therapy. In the future, elucidating the role of GPI-80 expression in the myeloid cells in tumor immunity and its manipulation may help improve the efficacy of tumor immunotherapy.

## Figures and Tables

**Figure 1 biomedicines-11-01296-f001:**
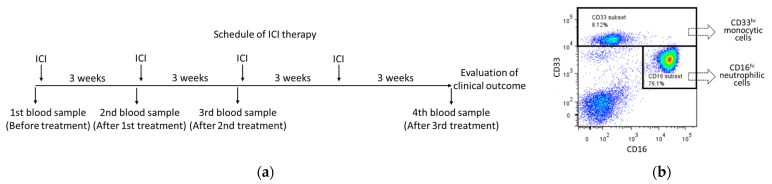
Collection and analysis of peripheral blood during ICI therapy. (**a**) Schedule of blood sample collection immediately before and during ICI therapy. Blood was collected immediately before each ICI therapy session, and clinical outcomes were evaluated using RECIST version 1.1 after the end of treatment. (**b**) Representative analysis of neutrophilic and monocytic cell populations. Whole blood cells were stained with CD33, CD16, GPI-80, and LAP-1, as described previously [10]. The cell populations of CD33^hi^ monocytic cell type or CD16^hi^ neutrophilic cell type were gated.

**Figure 2 biomedicines-11-01296-f002:**
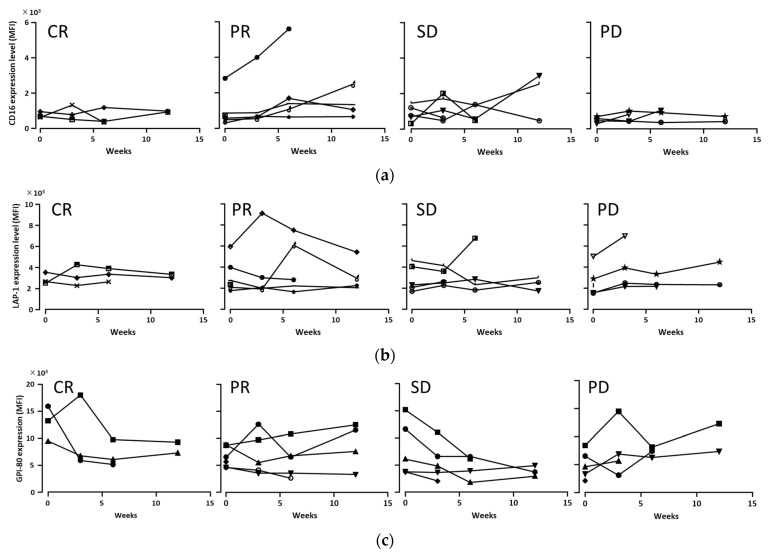
Variable patterns of myeloid cell parameters (CD16, LAP-1, and GPI-80) before and during ICI therapy. The patients treated with ICI therapy as the 1st line were separated into four categories as shown in Table 2: complete response (CR; *n* = 3), partial response (PR; *n* = 6), stable disease (SD; *n* = 5), and progressive disease (PD; *n* = 5). The changing of the mean of fluorescence intensity (MFI) of CD16 (**a**) and LAP-1 (**b**) in CD33^hi^ monocytic cells or the MFI of GPI-80 (**c**) in CD16^hi^ neutrophilic cells were measured by flow cytometry using blood samples collected from the patients during ICI therapy.

**Figure 3 biomedicines-11-01296-f003:**
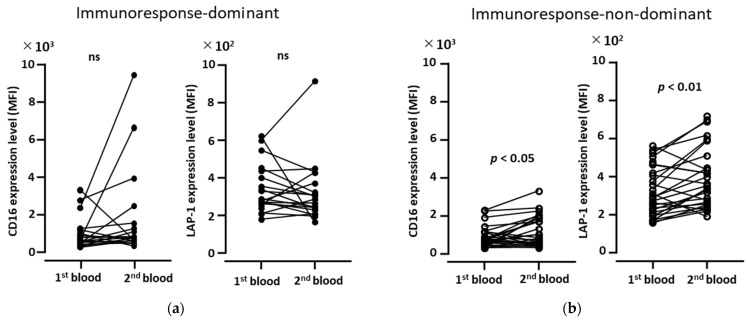
Increased expression of CD16 and LAP-1 in CD33^hi^ monocytic cells in immunoresponse-non-dominant patients after the 1st treatment. In immunoresponse-dominant patients ((**a**); *n* = 18) and immunoresponse-non-dominant patients ((**b**); *n* = 26), the expression levels (MFI) of CD16 and LAP-1 in CD33^hi^ monocytic cells were measured using 1st and 2nd blood samples with flow cytometry. During the chemotherapies, 2nd blood samples were collected 2–3 weeks from the 1st blood sample collection, the same as for the ICI therapies. Statistical significances were calculated using the Wilcoxon rank test (two-tailed). Unpaired samples were removed from the data.

**Figure 4 biomedicines-11-01296-f004:**
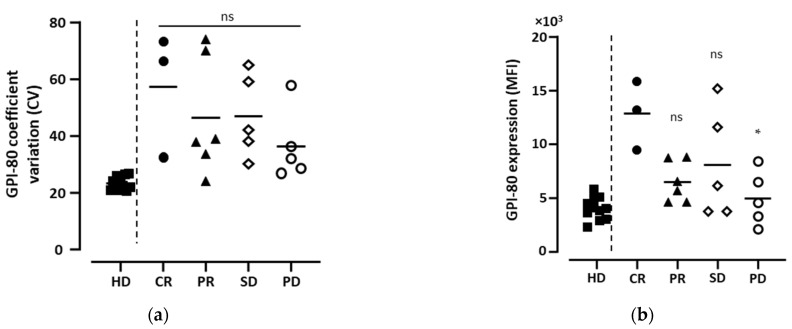
Associations of GPI-80 expression levels in CD16^hi^ neutrophilic cells immediately before ICI therapy with antitumor immune responses induced by ICI therapy. Both GPI-80 coefficient variation (CV; (**a**)) and GPI-80 MFI (**b**) in CD16^hi^ neutrophilic cells in blood samples were measured using flow cytometry. The blood samples were collected from healthy donors (HD represented by closed square; *n* = 11) or were the 1st blood samples (immediately before ICI therapy) of patients with RCC who received ICI therapy as the 1st line. The patients consisted of complete response (CR represented by closed circle; *n* = 3), partial response (PR represented by closed triangle; *n* = 6), stable disease (SD represented by open rhombus; *n* = 5), and progressive disease (PD represented by open circle; *n* = 5). The statistical significance was calculated using non-parametric ANOVA (Kruskal–Wallis) with a post-hoc test using Dunne, comparing CR vs. PR, SD, or PD. Each bar in figure is the mean of the data; ns—not significant, * *p* < 0.05.

**Figure 5 biomedicines-11-01296-f005:**
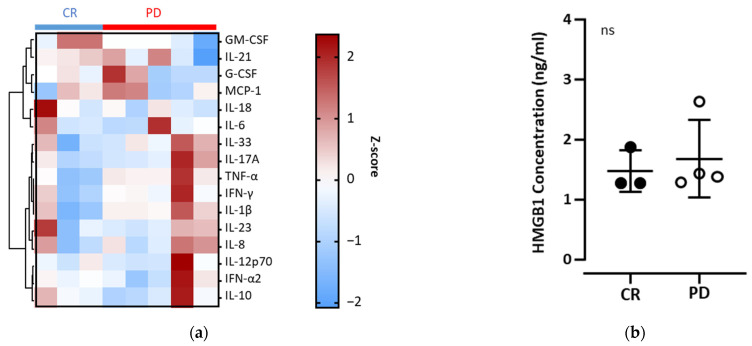
No relation of proinflammatory factors in plasma with clinical outcome. Plasma collected from 1st blood sample (before treatment) from patients with complete response (CR) and progressive disease (PD) with the same samples as Figure 4. (**a**) Heat map of cytokine profile. Cytokines in the plasma were measured by LEGENDplex™ Human Inflammation Panel 1 and then unsupervised clustering was analyzed using Z-score distribution with Prism software (CR, *n* = 3; PD, *n* = 5). (**b**) Comparison of HMGB1 concentrations between patients with CR and PD. HMGB1 in the plasma were measured by ELISA, and data were analyzed using two-tailed unpaired Student’s *t*-tests. Each datum is presented as a dot with the mean (bar) ± standard deviation (CR, *n* = 3; PD, *n* = 4; ns, not significant).

**Table 1 biomedicines-11-01296-t001:** Characteristics of patients with RCC in this study.

Total number of patients	N
	51
Age	median of years (range)
	69.0 (40–84)
Sex	n (%)
Male	42 (82.3)
Female	9 (17.7)
Histology	n (%)
Clear cell	39 (76.5)
Papillary	1 (1.9)
Bellini duct	1 (1.9)
Chromophobe	3 (5.9)
Acquired cystic disease-associated	1 (1.9)
Unclassified	6 (11.9)
^1^ T stage	n (%)
1a	3 (5.9)
1b	10 (19.7)
2a	9 (17.7)
2b	1 (1.9)
Unclassified	3 (5.9)
3a	23 (45.1)
3b	1 (1.9)
3c	1 (1.9)
^1^ N stage	n (%)
0	43 (84.3)
1	3 (5.9)
2	5 (9.8)
^1^ M stage	n (%)
0	2 (4.0)
1	49 (96.0)
Site of Metastasis	n (%)
Lung	35 (68.6)
Lymph node	19 (37.2)
Bone	18 (35.2)
Liver	6 (11.9)
Pancreas	4 (7.8)
Adrenal	4 (7.8)
Brain	4 (7.8)
Skin	1 (1.9)

^1^ Tumor-node-metastasis (TNM) stage was determined according to The Union for International Cancer Control (UICC), 2017.

**Table 2 biomedicines-11-01296-t002:** Patients’ clinical outcomes with therapy lines and these regimens.

Category	Regimen	Therapy Lines						Immunoresponse-Dominant		Immunoresponse-non-Dominant		NE
		1st	2nd	3rd	4th	5th	6th	CR	PR	SD	PD	
ICI	N + I	15						3	4	3	5	
	N		2	4	3				2	5	2	
ICI + TKI	A + A	3							2	1		
	P + A	1								1		
Chemotherapy												
	SO		2		1				1	1	1	
	CABO	1	5	3	4	2	1		8	6	1	1
	PAZ			1						1		
	AX			1						1		
	TS-1					1				1		
	EV				1					1		
n		20	9	9	9	3	1	3	17	21	9	1
(%)		39.1	17.7	17.7	17.7	5.9	1.9	5.9	33.3	41.2	17.7	1.9

Abbreviations: CR—complete response; PR—partial response; PD—progressive disease; SD—stable disease; NE—not evaluated; ICI—immune checkpoint inhibitor; TKI—tyrosine kinase inhibitor; N + I—Nivolumab plus Ipilimumab; N—Nivolumab; A + A—Avelumab plus Axitinib; P + A—Pembrolizumab plus Axitinib; SO—Sorafenib; CABO—Cabozantinib; PAZ—Pazopanib; AX—Axitinib; TS-1—Fluoropyrimidine class of chemo-therapeutic agent; and EV—Everolimus.

## Data Availability

The data presented in this study are available from the corresponding author upon request.

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
