# Peer review of "Initial Myeloid Cell Status Is Associated with Clinical Outcomes of Renal Cell Carcinoma"

_biomedicines, 2023, doi:10.3390/biomedicines11051296_

Round 1

Reviewer 1 Report

Myeloid-derived suppressor cells (MDSCs) strongly suppress tumor immunity. In this study, we investigated whether the clinical outcome of ICI therapy could be predicted by estimating the status of myeloid cells, including MDSCs. They found that immediately before ICI therapy, GPI-80 expression in neutrophils was significantly higher in patients with a complete response than in those with disease progression, which may indicate the differeation and maturation of a cluater of MDSCs. These results are interesting, suggesting that the status of myeloid cells during the initial phase of ICI therapy and clinical outcomes could be potential indicators of prognosis of RCC patients.

Author Response

Comments from Reviewer 1

Myeloid-derived suppressor cells (MDSCs) strongly suppress tumor immunity. In this study, we investigated whether the clinical outcome of ICI therapy could be predicted by estimating the status of myeloid cells, including MDSCs. They found that immediately before ICI therapy, GPI-80 expression in neutrophils was significantly higher in patients with a complete response than in those with disease progression, which may indicate the differeation and maturation of a cluater of MDSCs. These results are interesting, suggesting that the status of myeloid cells during the initial phase of ICI therapy and clinical outcomes could be potential indicators of prognosis of RCC patients.

Response to reviewer 1

Thank you for the time and effort that you have dedicated to providing your valuable feedback on our manuscript. We are grateful to you for your insightful comments on our paper and for recognizing our work. We are indebted for understanding the main goal of our manuscript and the implications of our method. 

Reviewer 2 Report

The article is devoted to an urgent topic, namely the search for markers of effective immunotherapy. It is worth noting that the studies were carried out at a high methodological level, however, the description of clinical data in the materials and methods is insufficient, there is no data on the histological subtype of the tumor. It is necessary to supplement the research material section and the discussion section, as interesting data have been obtained that require careful analysis

Author Response

Comments from Reviewer 2

The article is devoted to an urgent topic, namely the search for markers of effective immunotherapy. It is worth noting that the studies were carried out at a high methodological level, however, the description of clinical data in the materials and methods is insufficient, there is no data on the histological subtype of the tumor. It is necessary to supplement the research material section and the discussion section, as interesting data have been obtained that require careful analysis.

Response to reviewer 2

We appreciated the encouraging, critical and constructive comments on this manuscript by the reviewer. The comments have been very thorough and useful in improving the manuscript. We strongly believe that the comments and suggestions have increased the scientific value of revised manuscript by many folds. We have taken them fully into account in revision. We are submitting the corrected manuscript with the suggestion incorporated the manuscript. The manuscript has been revised as per the comments given by the reviewer as below:

  • We have mentioned the percentage of histological subtype in Table 1 of “Materials and Methods” section. As per reviewer’s suggestion, we added the following lines in “Result” section and added histological subtypes of CR, PR, SD, and PD in Table S1 in supplemental material as follows:

Result: As shown in Table S1, there was no association between the TNM grade and clinical outcomes in patients treated with ICI therapy as 1st line, while CR cases had histologically diverse RCC subtypes rather than PR, SD, and PD cases who had mostly clear cell RCC.

Table S1. TNM stage and histology of CR, PR, SD, and PD

Patients treated with ICI therapy at first line

n

19

Best response (n)

TNM stage

Histology

CR (3)

Case 1

T3aN0M1

Bellini duct

Case 2

T3aN1M1

Acquired cystic disease-associated

Case 3

T2aN0M1

Clear cell

PR (6)

Case 1

T3aN2M1

Clear cell

Case 2

T3aN0M1

Clear cell

Case 3

T3aN0M1

Clear cell

Case 4

T3aN0M1

Clear cell

Case 5

T3aN0M1

Clear cell

Case 6

T3aN0M0

Clear cell

SD (5)

Case 1

T3aN0M1

Clear cell

Case 2

T3bN0M1

Clear cell

Case 3

T3aN0M1

Clear cell

Case 4

T2N0M1

Clear cell

Case 5

T3aN2M1

Chromophobe

PD (5)

Case 1

T1b N0M1

Clear cell

Case 2

T1b N0M1

Clear cell

Case 3

T3cN0M1

Clear cell

Case 4

T3aN0M1

Clear cell

Case 5

T3aN0M1

Unclassified

  • We added the following discussion regarding the histological subtypes of RCC in the “Discussion” section:

RCC is a histologically heterogenous cancer with most common subtype clear cell RCC (ccRCC) and all subtypes other than clear cell categorized as non-clear cell RCC (nccRCC). In this study, only three cases of CR were reported with both ccRCC and nccRCC. nccRCCs have limited treatment options with the tendency of getting worse. Multiple clinical trials are now using ICI therapy in nccRCC, though the effectiveness of ICI therapy in nccRCC is not clearly understood yet [40]. Since patients with nccRCC showed better response toward ICI therapy in this study, we will continue the investigation with both histological subtypes.

Added reference

40. Zarrabi, K.; Walzer, E.; Zibelman, M. Immune Checkpoint Inhibition in Advanced Non-Clear Cell Renal Cell Carcinoma: Leveraging Success from Clear Cell Histology into New Opportunities. Cancers. 2021; 13. DOI: 10.3390/cancers13153652.
